# Splanchnic Vein Thrombosis in Myelofibrosis—An Underappreciated Hallmark of Disease Phenotype

**DOI:** 10.3390/ijms242115717

**Published:** 2023-10-29

**Authors:** Elina A. Beleva

**Affiliations:** 1Clinic of Hematology, Military Medical Academy, 1606 Sofia, Bulgaria; elina.beleva@biomed.bas.bg; 2QSAR and Molecular Modelling, Institute of Biophysics and Biomedical Engineering, Bulgarian Academy of Sciences, 1113 Sofia, Bulgaria

**Keywords:** myelofibrosis, venous thrombosis, splanchnic circulation, myeloproliferative disorders

## Abstract

Splanchnic vein thrombosis (SVT) encompasses thrombosis in the vessels of the splanchnic basin and has a relatively rare occurrence with a reported frequency in the general population of 1–2%. An episode of seemingly unprovoked SVT almost always triggers a diagnostic work-up for a Philadelphia chromosome-negative myeloproliferative neoplasm (MPN), since atypical site thrombosis is a hallmark of MPN-associated thrombophilia. Primary myelofibrosis (PMF) is a rare MPN with an estimated incidence between 0.1 and 1/100,000 per year. Although prothrombotic tendency in PMF is not envisioned as a subject of specific therapeutic management, unlike other MPNs, such as polycythemia vera (PV) and essential thrombocythemia (ET), thrombotic risk and SVT prevalence in PMF may be comparably high. Additionally, unlike PV and ET, SVT development in PMF may depend more on procoagulant mechanisms involving endothelium than on blood cell activation. Emerging results from registry data also suggest that PMF patients with SVT may exhibit lower risk and better prognosis, thus highlighting the need for better thrombotic risk stratification and identifying a subset of patients with potential benefit from antithrombotic prophylaxis. This review highlights specific epidemiological, pathogenetic, and clinical features pertinent to SVT in myelofibrosis.

## 1. Introduction

Philadelphia chromosome-negative MPNs are a group of clonal hematopoietic stem cell disorders comprising the main entities PV, ET, and PMF. PMF has an estimated incidence between 0.1 and 1/100,000 per year and an age peak between 60 and 70 years [1]. The phase exhaustion of the myelopoietic reserve and gradient deposition of reticulin and collagen fibrosis in the bone marrow characterizes PMF. Two subentities represent PMF: prefibrotic MF, which can mimic ET at onset, and overtly fibrotic MF [2]. Extramedullary hematopoiesis in the spleen and liver, pancytopenia, and leukemic transformation are clinical consequences of the natural disease evolution [3]. The common symptoms include fatigue, night sweats, low-grade fever, early satiety, weight loss, abdominal fullness or discomfort, dysuria, hematuria, gastrointestinal bleeding, arthralgia, and bone pain [4]. PMF patients also suffer thrombotic complications, which are attributed mainly to disease-induced hemostatic dysregulation. The average reported frequency of thrombosis at diagnosis of PMF is nearly 10%, which is still significantly higher than that of the general patient population [5]. Moreover, prefibrotic PMF has an increased rate of thrombotic events, which is similar to ET [6]. Considering that a proportion of thrombotic complications in MF may undergo subclinically or be masked by other clinical symptoms, the real incidence of thrombosis in PMF may be even higher. In light of this, the prothrombotic tendency in PMF represents a clinical challenge because thrombosis may further aggravate clinical condition, increase mortality risk, and compromise therapeutic success attained by administered targeted therapy. This review aims to outline specific epidemiological, pathogenetic, and clinical features of splanchnic thrombosis that are pertinent to PMF and provide considerations for unmet clinical needs of SVT in PMF.

## 2. Splanchnic Vein Thrombosis—General Considerations

SVT encompasses portal, splenic, mesenteric, or hepatic vein thrombosis. The reported frequency of SVT from population studies is 1–2% [7,8]. Risk factors may be local (abdominal), such as inflammatory bowel disease, pancreatitis, cirrhosis, surgery, celiac disease, and hepatobiliary cancer, or systemic, including myeloproliferative neoplasms, oral contraceptive use, connective tissue disorders, thrombophilia, and infection [9]. According to a population study based on 23,796 autopsies, the frequency distribution of risk factors is hepatobiliary malignancy—67%, cirrhosis—28%, abdominal infection/inflammation—10%, MPN—3%, and idiopathic—14% [8]. Most cases are “secondary” to either local or systemic causes, as an improvement of diagnostic capabilities has led to refinement in the identification of possible risk factors, and the frequency of idiopathic SVT has been reduced to about 15% [10]. Underlying etiology may be associated with a predilection to thrombosis of certain splanchnic regions. For example, local factors predominantly provoke PVT and MVT, whereas the most typical risk factors for BCS are systemic prothrombotic conditions [10].

Concerning the rate of symptom manifestation, SVT may take an acute or chronic course. Both forms may be considered different stages of the same temporal continuum [11]. The length of exposure to thrombotic risk factors, i.e., whether permanent/systemic (e.g., cancer, MPNs) or transient/local (e.g., abdominal surgery or infection), determines the rate of vascular occlusion. The development of a compensatory vascular collateral network is inversely related to the occlusive rate, and its presence defines the severity of the clinical course as acute or protracted (chronic). General clinical symptoms of the different SVT subtypes are abdominal pain of variable severity, hepatomegaly, splenomegaly, and non-specific gastrointestinal symptoms, such as nausea, vomiting, and diarrhea. Additionally, separate anatomical entities present with specific features: ascites is rather typical for BCS; portal hypertension, variceal bleeding, and hypersplenism are characteristic of PVT; whereas signs of intestinal ischemia, such as ileus, hematochezia, postprandial pain (mesenteric angina), and abdominal pain with lumbar irradiation, are more prominent symptoms in MVT [12,13,14]. SVT anatomical subtypes also exhibit variable preponderance to chronic or acute thrombo-embolic occlusion. While non-acute occlusion of hepatic veins can be found in only 15% of BCS cases, the proportion of chronic forms in PVT and MVT may amount to 40% for each [14,15,16]. Chronic and subacute forms are considered to prevail in MPNs because the endothelial procoagulant switch is the leading prothrombotic mechanism. However, the prevalence of chronic and subacute forms in MPN patients is currently unknown.

The prognosis of SVT is variable and depends on the acuity of clinical course, affected vascular network, predisposing factors and ability to tolerate anticoagulation. Estimated survival during the first post-thrombosis year is reported to be 82% (95% CI, 77–87) for BCS and 69% (95% CI, 61–76) for PVT [12,17]. Patients with MVT seem to have the worst outcome, as few small retrospective studies have reported a 30-day survival rate of around 20% [18]. Another risk factor for morbidity and long-term sequelae in patients after the first episode of SVT is recurrent thrombosis. A prospective study evaluating the outcomes of SVT in 604 patients has found a recurrent thrombosis rate of 7.3/100 patient years, and fatal outcome in 13.2% (95% CI 6.60–24.15) of them [19].

## 3. Methods

The literature was searched in a systematic manner using PubMed/MEDLINE from its inception to 30 July 2023. The search items were listed as follows: (1) “splanchnic”; (2) “mesenteric”; (3) “hepatic vein”; (4) “hepatic” AND “vein”; (5) “lienal”; (6) “portal vein”; (7) “portal” AND “vein”; (8) “splenic vein”; (9 “splenic” AND “vein”; (10) “Budd–Chiari”; (11) “thrombosis”; (12) “myelofibrosis”; (13) “primary myelofibrosis”; (14) “PMF”; (15) “myeloproliferative”; (16) “MPN” in [All Fields] queries. Each one of items 1–9 was search in combination with item 11 “thrombosis” AND with each one of items 12–16. For example: “splenic vein” AND “thrombosis” and “myelofibrosis”. “Budd-Chiari” item (5) was searched in combination with items 12–16 only, e.g., “Budd-Chiari” AND “myelofibrosis”. Search results were downloaded as CSV files and merged. Duplicates were removed by PubMed reference number, and articles not in English were excluded. After screening by title and abstract case reports, studies assessing only PV and ET patients or studies not on Ph-negative MPNs were excluded. The remaining 280 references underwent full-text review, and items were included if they provided relevant evidence-based data: reported MF-SVT cases separately, had MPN-SVT sample size ≥ 20, and had available full text. Additionally, references were screened to find other relevant articles. The final number of references totaled up to 92, as shown in Figure 1.

## 4. Epidemiology of SVT in Myelofibrosis

Diagnostic work-up for an MPN is almost always triggered after an episode of seemingly unprovoked SVT, as atypical site thrombosis is a hallmark of MPN-associated thrombophilia. The spectrum of occlusive events includes both arterial and venous thromboses, with arterial episodes occurring three times as often as venous ones [20]. The relative frequency of veno-occlusive episodes in MPN is between 0.5 and 1.3 per 100 patient years [21]. In about 40–70% of patients diagnosed with non-cirrhotic non-solid tumor-associated SVT, an underlying MPN is diagnosed. Guidelines recommend, therefore, testing for *JAK2*-V617F mutation and MPN diagnostic work-up in patients with unprovoked splanchnic thrombosis. A meta-analysis on the prevalence of MPN in primary non-cirrhotic non-malignant SVT found a 6.7% prevalence of MF (95% CI, 3.7–11.9) in BCS patients and 12.8% prevalence of MF (95% CI, 8.0–19.9) in PVT patients. In comparison, the prevalence of PV and ET was, respectively, 52.9% (95% CI 42.2–63.4) and 24.6% (95% CI 18–32.5) in PVT [22]. A similar rate was reported in an observational single-center study—4.2% prevalence of MF in BCS and 13.7% in PVT patients [23]. The prevalence rate of MF in newly diagnosed patients with unprovoked non-cirrhotic nonmalignant SVT seems to be lower than that of PV and ET.

In patients with diagnostically verified MPN, there is a heightened propensity to develop thrombosis in the splanchnic circulation. SVT prevalence rates range between 10 and 20% in MPN patients compared to 3.8/100,000 in the general population [24]. While SVT in PV and ET occurs at a relatively stable frequency as reported by various study cohorts 20–40%, reported rates of SVT prevalence in patients with MF are more dispersed and range between 6 and 37%, as shown in Table 1 [21,25,26,27,28]. This difference may partly be due to sampling imbalance, as almost all studies assessing SVT in MPN include predominantly patients with PV and ET as they have generally favorable disease course with smaller disease burden, and thrombotic complications constitute the major morbidity defining factors. A further extension is that assessing vascular risk factors and reducing thrombotic risk is the mainstay of PV and ET patient management, as outlined by professional guidelines. As disease biology in MF is drastically more aggressive, current prognostic scoring systems are based on variables representing disease burden (such as peripheral blast count, bone marrow failure, and constitutional symptoms). The treatment strategy aims to alleviate disease-related symptoms, not thrombotic risk [29]. While thrombotic risk assessment is not incorporated into management of patients with MF due to disease complexity and confounding effects of therapy, thrombosis and particularly SVT may occur at a similar rate in patients with MF. Additionally, typical symptoms in MF, such as splenomegaly due to extramedullary hematopoiesis and abdominal pain related to organ compression, may mask the classic symptoms of thrombosis in the splanchnic veins, thus leading to a lower suspicion rate for SVT in MF and underdiagnosis of SVT episodes. Some studies report SVT rates in MF patients as high as those in PV and ET [28,30]. In certain studies, SVT was significantly associated with MF, and cumulatively, primary and secondary MF accounted for as much as 50% of the SVT cases, as shown in Table 1 [27,31,32,33]. These data suggests that SVT in MF may be as prevalent as in PV and ET and impose the same morbidity burden, which, when added to the initially higher prognostic risk for these patients, may further aggravate prognosis. However, as MF seems to be underrepresented in almost all studies evaluating SVT in MPN patients, there is limited opportunity to extrapolate findings from those studies on MF given the differences in biology and prognosis.

## 5. Pathogenesis of SVT in Myelofibrosis

The mechanism of prothrombotic tendency in MF still needs to be fully elucidated. Much of our understanding comes from studies in ET and PV, and inferences are made in general for MPNs including MF. In the pathophysiology of MPN-associated SVT, several prothrombotic factors converge—activated blood cells, altered blood flow and pro-adherent endothelial phenotype—to increase thrombotic predisposition in the splanchnic basin, as shown in Figure 2. In MPN, hyperactive *JAK2* signaling has been implicated in the activation of blood cells, specifically by the expression of high levels of P-selectin, adhesive integrins, tissue factor, and engagement in neutrophil extracellular trap formation, leading to enhanced interaction with the endothelium [54,55,56,57,58]. Prolonged interaction between circulating cells and the endothelium is further enabled by the physiologically slow velocity of blood flow in the splanchnic vasculature. This can be enhanced by increased ICAM1, VCAM1, vWF and P-selectin expression by endothelial cells—a prothrombotic phenotype conferred particularly by *JAK2*-V617F mutation [59].

However, in MF, as suggested by data from translational studies, the splanchnic prothrombotic tendency may be less dependent on blood cell activation than it is in PV and ET, as shown in Figure 2. A study evaluating NETs formation in patients with MPNs showed that triggering NETosis by potent stimuli such as phorbol 12-myristate 13-acetate was impaired particularly in MF patients due to decreased reactive-oxygen species production by neutrophils [55]. A study that assessed platelet properties in a Vav1-h*JAK2*^V617F^ knocked-in mouse model of PMF found significantly delayed thrombus formation upon vascular injury because platelets had compromised ability to secrete adenosine diphosphate due to reduced collagen activation response and a diminished number of dense granules [60]. In comparison, in a mouse model of ET, *JAK2*-V617F platelets showed enhanced platelet reactivity and aggregation [61].

On the other hand, a few studies have explored the role of endothelial cell dysfunction for thrombosis formation in MPN. As suggested by their findings, MF may owe its prothrombotic tendency to procoagulant conversion of the endothelium. EDA-FN, a marker of endothelial activation and thrombo-inflammation, was elevated in PMF patients while undetectable in healthy controls. Levels of EDA-FN were significantly higher in PMF-SVT patients compared to PMF patients without SVT [62]. Soluble thrombomodulin, another marker suggestive of continuous endothelial damage, was increased in the plasma of PMF patients vs. healthy controls. This difference was significant irrespective of the patient’s *JAK2*-V617F status [63]. Increased circulating endothelial, erythrocyte, platelet and tissue factor microparticles were found in MPN patients with thrombosis vs. non-thrombosis [64]. This study observed the highest counts of all microparticle subpopulations in the PMF subgroup. Circulating endothelial cells have been considered biomarkers for vascular injury and thrombotic risk. Their count correlated inversely with the degree of thrombosis recanalization in MPN patients with SVT (50% were MF patients) [65]. Another study demonstrated that *JAK2*-V617F mutated E-CFCs exhibited higher adhesion proficiency to mononuclear cells than normal E-CFCs. E-CFC levels were higher in MPN patients vs. healthy controls with this difference being significant only for the PMF patient group [66]. Rosti et al. found that an increased frequency of E-CFCs was predictive of a history of SVT in the MPN patient population enriched in low and intermediate-1 risk MF cases [38]. In the same study, the highest numbers of E-CFCs were observed in patients with prePMF. Thus, the ability to mobilize E-CFCs is correlated with increased thrombotic response in PMF.

## 6. Predictive Risk Factors and Clinical Outcomes of SVT in Myelofibrosis

Although it has been acknowledged that the incidence of vascular events in MF may be comparable to that of PV and ET, there is limited knowledge on the factors conferring risk in MF and the usefulness of preventive strategies. Moreover, the substantial bleeding risk and heterogeneity of the MF patient population in terms of morbidity further render risk assessment difficult. Several clinical risk factors have been cited as all-cause thrombosis risk factors in MPN [67]. Age, previous thrombosis and cardiovascular risk factors have been incorporated in the PV and ET two-tiered models for thrombosis and IPSET [29,68]. The IPSET thrombosis model was developed to estimate thrombotic risk in newly diagnosed patients with ET based on age, previous thrombosis, cardiovascular risk factors and *JAK2*-V617F status [68]. Recently, IPSET has been validated in patients with prePMF as an effort to provide for the first time a score that may be informative for thrombosis risk in this elusive to manage clinical condition [6]. Results from this validation study on 382 prePMF patients indicate that only a history of previous thrombosis was significantly predictive of venous thromboembolism in prePMF (HR 3.06, 95% CI 1.41–6.4, *p* = 0.005), and this was particularly relevant if the previous event was of venous origin (HR 5.53, 95% CI 2.32–12.2, *p* < 0.0001). In comparison, IPSET risk factors such as age > 60, leukocyte count > 10 G/l and the presence of at least one cardiovascular risk factor (history of smoking, hypertension, diabetes mellitus, hypercholesterolemia) significantly conferred increased risk for arterial thrombotic events in prePMF similarly to ET. Interestingly, SVT accounted for nearly two-thirds (68%) of all venous events at diagnosis and one-third (29%) at follow-up. The finding that established risk factors for thrombosis in ET are predictive of arterial events in preMF but not of venous ones, represented in the majority by SVT, provides clinical support to the assumption that other risk factors and, hence, pathogenetic mechanisms play a role in SVT development in MF.

An association between IPSS and thrombosis has been observed in a few studies. A large registry-based study from Spain identified IPSS as independently associated with an increased incidence of thrombosis during the follow-up of MF patients [69]. After adjustment for antithrombotic and main cytoreductive treatments, patients with intermediate-2/high IPSS had a two-fold greater risk for thrombosis during follow-up than the lower-risk groups. Similarly, in another population-based study, higher IPSS categories were significantly more often observed among MF cases with vascular complications (52%) than matched MF controls (33%). However, no significant difference was found in the frequency distribution according to the dynamic international prognostic scoring system [70]. The Myelofibrosis Secondary to PV and ET—Prognostic Model also did not show potential for predicting thrombosis in 680 patients with secondary MF [69]. It is important to note, however, that both of these studies included arterial and venous vascular events together in the multivariate analysis to estimate the risk factors for thrombosis. Somewhat contrasting to the results reported from the studies mentioned above are findings from studies on selected MPN patients based on the occurrence of SVT. In an international retrospective study, 518 cases with MPN-SVT were compared to 1628 unselected control MPN patients matched for disease subtype. In the MPN-SVT group, the PMF-SVT subgroup emerged with a distinct clinical phenotype: SVT cases were significantly enriched in PMF with lower IPSS and PMF-SVT cases had longer OS than PMF controls [35]. Findings were consistent also for the prePMF subgroup. Even though this echoes already known distinctive clinical features of MPN-SVT patients such as female sex and younger age, a recent study from the framework of the ERNEST registry project sheds light on the potential interaction of increased thrombotic risk in *JAK2*-V617F-positive PMF patients with lower IPSS scores [71]. In this prospective study of 585 PMF patients, factors that significantly distinguished patients experiencing thrombosis post-diagnosis of PMF in the univariate model were younger age, lower IPSS and *JAK2*-V617F mutation. The combination of *JAK2*-V617F and lower IPSS was independently associated with thrombosis in the multivariate model when adjusted for previous thrombosis, hemoglobin levels and cytoreductive therapy. Furthermore, in *JAK2*-V617F-negative patients, the likelihood of thrombosis after ten years tended to be higher in the lower than in the higher IPSS risk category after adjusting for competing risk.

Altogether, these data highlight that thrombotic risk in MF patients is increased and that the IPSS low-risk category is associated with a heightened propensity to SVT. This necessitates better thrombotic risk stratification, particularly in low-risk PMF patients, and the consideration of antithrombotic prophylaxis in a subset of them.

## 7. Molecular Profiling and Thrombotic Risk in Myelofibrosis

Improving the knowledge of the association between somatic mutations and risk factors for thrombosis has been identified as an unmet clinical need in the management of MPN-associated thrombosis in a recent consensus-based paper [72]. As per the panel report statement, not only driver but also selected non-driver mutations may be implicated in the increased risk of thrombosis, and their predictive value needs to be evaluated in a prospective study as the amount of evidence comes from retrospective analyses. *JAK2*-V617F mutation and variant allele frequency have been almost unequivocally associated with increased thrombosis risk in MPN patients and in the general population [22,73,74,75,76]. In PMF, a significantly higher proportion of *JAK2*-V617F-positive status was found among patients with vascular complications (74%) than in those without (55%, *p* = 0.018) [70]. *JAK2*-V617F positive carrier status was also independently associated with two times higher thrombosis incidence during the follow-up of PMF patients irrespective of introduced preventive antithrombotic strategies, and the interaction between *JAK2*-V617F and leukocytosis further potentiated this effect [69,77]. In terms of survival outcomes, *JAK2*-V617F has been shown to negatively impact OS and TFS in prePMF compared to ET in a cohort of Taiwanese patients. In this study, frequencies of the three driver mutations and *CALR* allele burden were not different between prePMF and ET, but *JAK2*-V617F allele burden was significantly higher in prePMF. Furthermore, prePMF patients had a significantly elevated incidence of SVT than ET patients, and prePMF patients with *JAK2*-V617F had significantly inferior OS and TFS than prePMF patients with *CALR* mutations [73]. *JAK2* exon12 has been identified by NGS with a 1.3% prevalence in idiopathic non-cirrhotic SVT and with 40% prevalence (two out of five cases) in triple-negative MPN-SVT, further strengthening the site-specific association of *JAK2* mutations with SVT [78]. As opposed to the prothrombotic influence of *JAK2* mutations, *CALR* and *MPL* driver mutations are reported with very low to almost absent prevalence in MPN-SVT (0.01–0.06% for *MPL*, 1–5% for *CALR*) [22,25,35,79]. Data on distribution frequency and the potential impact of *CALR* and *MPL* mutations in PMF-SVT patients are currently missing.

An important association between myeloid somatic non-driver mutations and SVT emerges from molecular profiling studies in splanchnic thrombosis. In a study by Magaz et al., NGS identified HMR variants associated with CHIP in 37.8% of 74 cases with idiopathic/local non-cirrhotic SVT (MPN excluded) [78]. *TET2*, *DNMT3*, *ASXL1*, *CEBPA*, and *CSF3R* were among the most frequent ones, and patients harboring HMR variants exhibited a significantly higher incidence of re-thrombosis even after adjustment for patients’ age. Another study reported an even higher prevalence of CHIP (46%) in a series of idiopathic SVT patients, further alluding to a potential role of clonal hematopoiesis in SVT development [80]. In two other MPN-SVT studies, a similar proportion of patients (around one-third) was reported to harbor high-risk genomic features as identified by NGS [25,31]. The study by Debureaux et al. was looking at the predictive capability of HMR status with respect to the primary outcome defined as the incidence of transformation to secondary myelofibrosis, acute leukemia or death, but it was not designed to model differences between HMR-negative vs. -positive status with respect to thrombosis [25]. Indeed, as expected, HMR-positive status was associated with significantly worse primary outcomes and shorter event-free survival and OS. On the other hand, the study by Cattaneo et al. explicitly looked for associations of HMR status with SVT recurrence or thrombo-hemorrhagic complications in a group of 58 consecutive MPN-SVT patients [31]. In this study, contrary to the observed associations between CHIP and SVT in the study of idiopathic SVT by Magaz et al., HMR features such as bi-allelic TP53 loss, *ASXL1* and *CSMD1* mutations were predictive of leukemic evolution, but neither were associated with SVT recurrence nor with thrombo-hemorrhagic complications. Further adding to the confusion are results from the IPSET validation study in prePMF, as HMR status was found to significantly increase the predictive power of IPSET in prePMF [6]. Still, the association of HMR status with thrombosis was only significant for arterial episodes not venous ones. These data confirm the predictive ability of HMR status for leukemic transformation and major survival outcomes also for the subset of patients with MPN-SVT. Whether the presence of high-risk genomic features also confers increased thrombotic risk, particularly in PMF patients, needs to be further established due to varying study results and the low number of patients with HMR and concurrent thrombotic events.

## 8. Treatment of SVT in Myelofibrosis

### 8.1. Role of Non-MF Directed Therapy

#### 8.1.1. Medical Treatment

There is limited evidence to guide the specific management of MPN-SVT, and data on MF-SVT are even scarcer or coming from single cases. In the acute setting, the management of MPN-SVT does not differ from the standard approach in non-cirrhotic SVT. Anticoagulation is the cornerstone and is generally the first line of antithrombotic therapy directed at achieving recanalization and maintaining vessel patency [81]. For both MPN-BCS and MPN-PVT, the approach follows general guidelines and constitutes anticoagulation with LMWH at a therapeutic dose and a subsequent switch to vitamin K antagonists [82,83]. In the Mayo Clinic experience, nearly one-third of the 29 patients with MF-SVT did not receive anticoagulation, another 31% (*n* = 9) received only systemic anticoagulation, and 38% (*n* = 12) received systemic anticoagulation with cytoreduction [28]. Death by liver failure (3 cases) and gastrointestinal bleeding (1 patient) was observed only in MF-SVT patients after a median follow-up of 2.7 years. While TFS was comparable among the three MPN subtypes, SVT-related unfavorable long-term outcomes seemed more prevalent in the MF-SVT subgroup, which had the highest proportion of patients not receiving anticoagulation.

Few studies also allude to the possibility that MPN-SVT patients fail to achieve a sufficient recanalization rate on anticoagulation. A complete recanalization rate after medical treatment was seen in 0 to only 8% of patients with MPN-SVT, and *JAK2*-V617F was associated with 76% greater odds of failure to recanalize [40,84]. Given the presence of splenomegaly in the majority of MPN patients, particularly MF, failure to recanalize may heighten the risk of complications such as portal hypertension and variceal bleeding. This raises the question of whether early interventional treatment as an adjunct to anticoagulation may be beneficial to a certain subset of MPN-SVT patients. However, these studies have assessed MPNs together, so assumptions on MF-SVT can only be inferential.

In addition to LMWH and vitamin K antagonists, the use of DOACs in MPN-SVT is not firmly established. In a retrospective cohort of 64 MPN-SVT patients (15 MF patients), DOACs appeared to be safe in the treatment of SVT [36]. The MPN-DOACs study, however, found that among MPN patients receiving DOACs for either atrial fibrillation or venous thromboembolism incl. SVT, the proportion of patients who bled was significantly higher in MF than in other MPN subtypes [85]. In multivariate analysis, MF diagnosis retained its significance for bleeding on DOACs even when corrected for age, sex or indication for DOACs prescription.

#### 8.1.2. Interventional Treatment

A subset of patients with MPN-SVT may require the placement of stents in the abdominal veins, thrombolysis, or even OLT [86]. Data are scant, and the outcome of interventional procedures has not been studied systematically in MF. The decision to undergo invasive treatment shall be guided by the severity, rate of clinical deterioration, and presence of complications such as variceal bleeding and liver dysfunction. In one study of portosystemic shunt surgery in MF, the procedure was associated with the resolution of portal hypertension complications in 9 out of 10 MF patients [87]. Splenorenal shunt was performed in eight patients. The one patient in whom a portocaval shunt was performed died of sepsis as a post-intervention complication. In the remaining nine patients, survival ranged between 3 months and 12 years. In the Mayo Clinic MPN-SVT series, TIPS was placed in 17% of MF-SVT patients (*n* = 5), which was comparable to the other MPN subtypes [28]. In another series of 15 patients with TIPS placement for the treatment of BCS, the four patients with MF achieved significant portal decompression at day 60 post-intervention and maintained patent TIPS during follow-up. Only one of them required percutaneous transluminal angioplasty four months after TIPS placement.

Data on OLT for BCS in MF patients are restricted to single cases. In a study of the long-term follow-up (over 30 years) after OLT for BCS, MPN patients had approximately three times higher mortality risk. MPN patients constituted half of the studied population, but there was only one MF patient. This patient suffered splenic rupture five years after OLT, and pathologic examination revealed leukemic transformation [50]. In another study, out of 41 MPN-SVT patients who received OLT for BCS, only one MF was present in the series and had reported survival greater than three years [88].

### 8.2. Role of MF Directed Therapy

Based on current recommendations, the risk stratification guides MF-directed therapy. Cytoreduction with hydroxyurea may be an option in low and intermediate-risk symptomatic patients if deemed beneficial. Even though there is consensus on the use of cytoreduction with hydroxyurea in high-risk PV and ET patients, its antithrombotic efficacy in SVT is debatable. Experimental data suggest that hydroxyurea can reduce thrombosis by decreasing leukocyte rolling and adhesion in mesenteric venules after TNF-alpha administration [89]. These experimental observations align with findings from clinical studies on reduced thrombotic occurrence in post-PV and post-ET MF patients receiving cytoreductive therapy [90]. On the other hand, in the Mayo Clinic experience, post-SVT thrombosis survival and recurrence were not affected by the type of initial treatment strategy, incl. cytoreductive therapy with or without anticoagulation [28]. Hydroxyurea was not able to prevent recurrent thrombosis in the splanchnic region in a multicenter cohort of MPN patients [91]. Considering that SVT pathogenetically is related to enhanced *JAK2* signaling and given hydroxyurea’s lack of anti-inflammatory properties, its antithrombotic efficacy in SVT remains to be assessed.

The only *JAK2* inhibitor that has so far been evaluated in the context of SVT is ruxolitinib. In their work, Pieri et al. demonstrated that ruxolitinib was safe and well tolerated in MF-SVT patients [33]. Notably, after four weeks of treatment, levels of angiogenic and pro-inflammatory cytokines VEGF and soluble TNF decreased significantly as compared to baseline. A similar trend was observed for the levels of endothelial progenitor cells after 24 weeks of treatment [33]. Stationing of the grade of esophageal varices upon ruxolitinib treatment was seen in an observational study of MPN patients with portal hypertension. In 90% of patients, ruxolitinib was taken together with vitamin K antagonists. The extension of SVT has remained stable compared to baseline, and eight out of nine patients did not show worsening of esophageal varices. In a meta-analysis, Samuelson et al. found a significant reduction in thrombotic events in the ruxolitinib group for both arterial and venous thrombosis [92]. Moreover, the highest values of vessel recanalization were observed in patients with PMF treated with ruxolitinib [65]. Even though the small number of studied patients limits these data, they provide clues for possible antithrombotic properties of ruxolitinib.

## 9. Future Directions and Conclusions

Given the relative paucity of data on SVT in PMF, several further investigations seem feasible. First, the prevalence of subclinical forms of SVT that may be masked by other clinical symptoms in PMF needs to be evaluated as this may unravel an unexpectedly high frequency of clinically significant but non-overt forms. Second, in a subset of PMF patients with low IPSS risk, thrombotic tendency may represent the major morbidity factor similar to ET and PV. To identify patients at heightened thrombotic risk and offer adequate prophylaxis, further research is needed on SVT risk factors as well as on the elucidation of prothrombotic mechanisms, and specifically, information on the role of endothelium is needed. Next, identifying HMR mutations by NGS and assessing their potential contribution to thrombosis presents an avenue for future research, as this may help further personalize antithrombotic management coupled with disease modification in certain patients. Last but not least, research on the antithrombotic properties of JAK2 inhibitor therapy is further warranted.

## Figures and Tables

**Figure 1 ijms-24-15717-f001:**
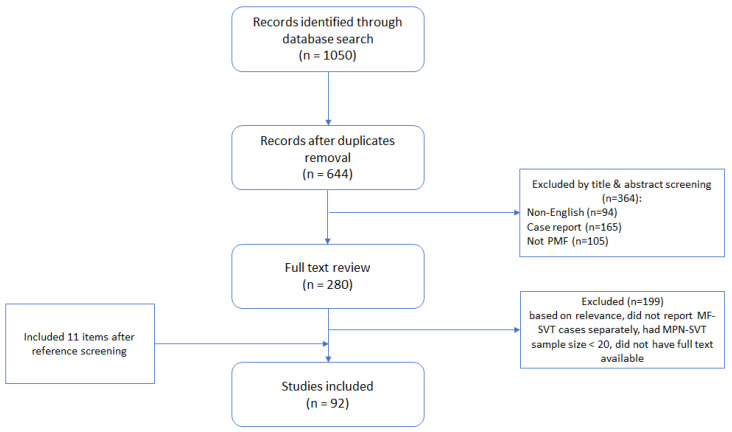
PRISMA flow chart of references selection.

**Figure 2 ijms-24-15717-f002:**
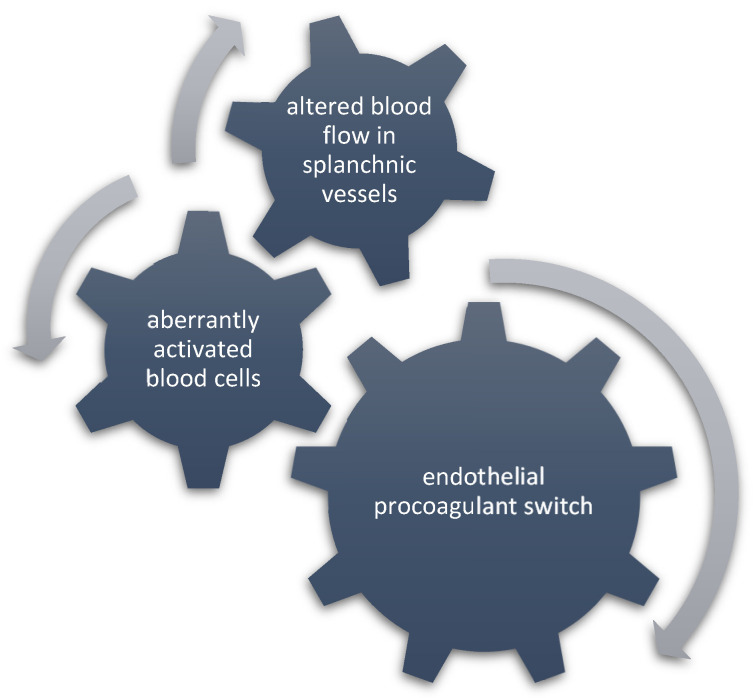
The relative contribution of Virchow’s triad components in MF-SVT development. An endothelial procoagulant switch may be the leading prothrombotic mechanism for SVT development in MF.

**Table 1 ijms-24-15717-t001:** Reported SVT prevalence in MPN subtypes.

Study	All MPN-SVT, *n* (100%)	PV, *n* (%)	ET, *n* (%)	MF, *n* (%)		MPNu, *n* (%)
De Stefano et al. [30]	181	67 (37)	67 (37)	47 (26)		NA
Kaifie et al. [27]	22	3 (13.63)	6 (27.3)	6 (27.3)	primary	3
				4 (18.2)	post-PV	
Lavu et al. [28]	84	29 (35)	26 (30)	29 (35)		NA
How et al. [34]	52	21 (41)	17 (33)	7 (13)		7 (13)
Sant’Antonio et al. [35]	518	192 (37)	178 (34.3)	68 (13)	primary	55 (11)
				20 (3.9)	prePMF	
				4 (0.7)	post-PV	
				1 (0.1)	post-ET	
Tremblay et al. [36]	64	29 (45)	14 (22)	8 (13)	primary	6 (9)
				2 (3)	prePMF	
				5 (8)	post-ET/PV	
Debureaux et al. [25]	80	52 (65)	23 (29)	5 (6)		NA
Görtzen et al. [37]	33	7 (21)	6 (18)	13 (40)		7 (21)
Gianelli et al. [32]	29	11 (37.9)	6 (20.6)	11 (37.9)		1 (3.4)
Cattaneo et al. [31]	58	9 (15.5)	8 (13.8)	4 (6.9)	primary	
				16 (27.6)	prePMF	16 (27.5)
				5 (8.6)	secondary	
Rosti et al. [38]	214	38 (17.7)	21 (9.8)	106 (49.5)	primary	NA
				49 (22.8)	prePMF	
Gonzales-Montero et al. [39]	26	5 (19.2)	12 (46.1)	4 (15.3)		
Naymagon et al. [40]	23	8 (34.7)	11 (47.8)	2 (8.7)		2 (8.7)
Fan et al. [23]	126	23 (18.2)	50 (39.6)	15 (12)		38 (30)
Ho et al. [41]	26	11 (42)	8 (30.7)	2 (7.7)	primary	3 (11.5)
				2 (7.7)	prePMF	
Pieri et al. [33]	21	5 (23.8)	4 (19)	8 (38.1)	primary	NA
				3 (14.3)	post-PV	
				1 (4.8)	post-ET	
Poisson et al. [42]	74	32 (43.2)	23 (31)	6 (8.1)		13 (17.5)
Yan et al. [43]	28	17 (60.7)	2 (7)	9 (32.1)		NA
Colaizzo et al. [44]	28	9 (32.1)	7 (25)	12 (42.8)		NA
Villani et al. [45]	108	NA	32 (29.6)	29 (26.8)	primary	21 (19.4)
				26 (24)	prePMF	
Smalberg et al. [46]	66	27 (41)	17 (25.8)	6 (9)		16 (24.2)
Janssen et al. [47]	23	12 (52.1)	3 (13)	6 (26)		2 (8.6)
Ollivier-Hourmand et al. [48] ^ω^	72	44 (61)	20 (27.7)	1 (1.3)		7 (9.7)
Darwish Murad et al. [49] ^ω^	49 (100)	27 (55)	9 (18.3)	2 (4)		11 (22.4)
Ibach et al. [50] ^ω^	22 (100)	5 (22)	9 (46)	1 (5)		7 (31)
Hoekstra et al. [51] ^Δ^	44 (100)	14 (31.8)	12 (27.2)	7 (15.9)		11 (25)
Primignani et al. [52] ^Δ^	23 (100)	3 (13)	14 (60.8)	2 (8.7)		4 (17.3)
Sahin et al. [53]	32 (100)	11 (34.3)	12 (37.5)	9 (28.1)		NA

^ω^ BCS only, ^Δ^ PVT only, *n*—number of patients, MPNu—MPN unclassifiable, NA—not applicable.

## Data Availability

Not applicable.

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
