# Peer review of "Splanchnic Vein Thrombosis in Myelofibrosis—An Underappreciated Hallmark of Disease Phenotype"

_ijms, 2023, doi:10.3390/ijms242115717_

Round 1

Reviewer 1 Report (Previous Reviewer 1)

The author has addressed all of my concerns and in my opinion the paper has been significantly improved. I would like to thanks the author on detailed revisions. I have no further comment and would recommend acceptance of the paper in the current form. 

n/a

Author Response

We thank the reviewer for taking time to review the manuscript and provide constructive comments. 

Reviewer 2 Report (New Reviewer)

Minor concerns need to be addressed.

Few typographical mistakes should be rectified.

Author Response

We would like to thank the reviewer once again for their detailed review and feedback. We have aimed at correcting the raised issues. Point-by-point answer is provided in the attached file. 

Reviewer 3 Report (New Reviewer)

Thank you for your work!

The author took into account the comments of the reviewers and corrected the text accordingly. The review provided by the author may be published after 1 minor correction.

- Figure 2 (line 210): The text should contain a link to a figure next to the text that explains it in some way. You made a drawing that figuratively explains the interaction of some phenomena, but nowhere is an explanation of this diagram given. In the caption to the picture, provide the statement on the basis of which you depicted it this way.

Author Response

We thank the reviewer for their input and time spent on reviewing the manuscript. We have added link to Figure 2 in line 202 and in line 212. In the caption of Figure 2 additional sentence has been added: "Endothelial procoagulant switch may be the leading prothrombotic mechanism for SVT development in MF."

This manuscript is a resubmission of an earlier submission. The following is a list of the peer review reports and author responses from that submission.

Round 1

Reviewer 1 Report

This is a very comprehensive review article on the topic of splanchnic vein thrombosis in patients with myelofibrosis. The review is quite comprehensive but there is a lot of repetition of information which makes it eventually boring to read. So, I would ask author to be more concise and to delete repetition. 

The table with main message should be created as well as graphical abstract so that reader can easily understand the main points of this research. 

Since the author is not an expert in this field, methodology section with detailed description how the literature was selected must be reported. Without this rigorous methodology section this review is not acceptable for publication. Please report how you selected references that were included in this article. What database you used, what key search words etc. Please take a look at prisma diagram. 

Specific comments: 

1. Line 69, please add celiac disease as a risk factor for portal vein thrombosis and superior mesenteric vein thrombosis ( Celiac Disease and Thrombotic Events: Systematic Review of Published Cases - PubMed (nih.gov))

2. Line 89- acute thrombotic occlusion should be changed to acute embolic occlusion. Furthermore, it should be emphasized that in chronic mesenteric occlusion, post prandial pain (mesenteric angina) might lead to weight loss and cachexia. 

3. In section 2 under general consideration- the outcome of thrombotic events also depends on ability and tolerance of anticoagulation. The author failed to mention this. 

4. In section 3 please compare rates of SVT in patients who have intraabdominal infection (suppurative portal vein thrombosis) and those who have malignancy related SVT). These are two distinct entities with different prognosis. Comparing just cirrhotic vs non cirrhotic group is not sufficient. 

5. Pathophysiology section is informative, but it is too long, please cut down word count and be more concise. 

6. Line 240 please explain IPSET

7. Treatment section is missing particularly if thrombus is infected (suppurative portal vein thrombosis) and utility of antibiotic therapy and more importantly anticoagulation choice and complications. 

minor editing needed, overall acceptable.

Reviewer 2 Report

The article entitled “Splanchnic Vein Thrombosis in Myelofibrosis – An Underappreciated Hallmark of Disease Phenotype” is a review that focuses on an clinical very important topic such as the trombotic risk in primary myelofibrosis (PMF). PMF bears the worst prognosis with a median overall survival of 4.4 years. In literature there are many studies about the transformation into acute myeloid leukemia but there are little or no articles on thrombosis in PMF. Therefore, this review is welcome. This article is well-written in conceptual and structural terms. The informations are comprehensive and the concepts are clearly state and easy to read. I have no observations to make. Therefore, I think that this article is suitable for publication in its current version.

Reviewer 3 Report

In the submitted manuscript the author focuses on the occurrence of splanchnic vein thrombosis (SVT) in patients with primary myelofibrosis (PMF). The topic is important as the current PMF treatment guidelines do not include a specific therapeutic management of thrombosis. The review is well written covering the epidemiology, clinical picture, pathogenesis and predictive risk factors of thrombotic events, in particular SVT, in PMF.

However, there are some suggestions to improve the scientific content of the paper:

1.     Abstract: Include the aim of the review.

In the statement “Primary myelofibrosis (PMF) is a rare MPN with estimated annual incidence between 0.1-1/100 000 per year” remove the word “annual”.

2. Introduction: At the beginning, mention the three entities of the classical Philadelphia negative myeloproliferative neoplasms. Also, introduce the two categories of PMF: prefibrotic and overtly fibrotic, highlighting that prefibrotic PMF can mimic essential thrombocythemia (ET) at onset, both entities being characterized by similarly increased rate of thrombotic events (Guglielmelli et al, Blood Cancer J., 2020).

 The phrase “The common symptoms include fatigue, night sweats, low-grade fever, early satiety, weight loss, abdominal fullness or discomfort, dysuria, hematuria, gastrointestinal bleeding, arthralgia, bony pain” does not have a reference.

Reference no. 3 indicates the prevalence of thrombosis at diagnosis of MPN – this is not mentioned in the manuscript! Also, the statement “Reported incidence of vaso-occlusive complications commonly to the MPN group is between 20-33%” is not found at reference no. 3.

3.    Predictive risk factors and clinical outcomes of SVT in myelofibrosis

Please, reformulate the phrase as it sounds too complicated: “Altogether these data have several implications: there is increased thrombotic risk in patients with MF, low-risk IPSS patient population may be at an increased risk of SVT and heightened clinical suspicion for SVT may be warranted as prophylactic anti-thrombotic treatment may be justified in a subset of low risk PMF patients”.

4.     Gene names should be italicized!

There are many punctuation mistakes that need correction.